# No Genus-Specific Gene Is Essential for the Replication of Fowl Adenovirus 4 in Chicken LMH Cells

Xinglong Liu,[a,b] Xiaohui Zou,[b] Wenfeng Zhang,[b,c] Xiaojuan Guo,[b] Min Wang,[b] Yingtao Lv,[a] Tao Hung,[b] Zhuozhuang Lu[b,d]

[a]College of Marine Science and Biological Engineering, Qingdao University of Science and Technology, Qingdao, Shandong, China
[b]State Key Laboratory of Infectious Disease Prevention and Control, National Institute for Viral Disease Control and Prevention, Chinese Center for Disease Control and Prevention, Beijing, China
[c]School of Laboratory Medicine, Weifang Medical University, Weifang, Shandong, China
[d]Chinese Center for Disease Control and Prevention–Wuhan Institute of Virology, Chinese Academy of Sciences Joint Research Center for Emerging Infectious Diseases and Biosafety, Wuhan, Hubei, China

Xinglong Liu and Xiaohui Zou contributed equally to this article. Author order was determined by drawing straws.

**ABSTRACT** Essential genus-specific genes have not been discovered for fowl adenovirus (FAdV), which hampers the development of FAdV-based vectors and attenuated FAdV vaccines. Reverse genetics approaches were employed to construct FAdV-4 mutants carrying deletions or frameshift mutations covering the whole left and right ends of the viral genome. The results of virus rescue and plaque forming experiments illustrated that all the 22 designated ORFs (open reading frames) were dispensable for the replication of FAdV-4 in chicken hepatoma Leghorn male hepatoma (LMH) cells and primary embryo hepatocytes. RNA-seq data demonstrated that ORF28 and ORF29 were not protein-encoding genes, and suggested a promoter (RP1) and an intron in these regions, respectively. The promoter activity of RP1 was further confirmed by reporter gene expression experiments. GAM-1-deleted FAdV-4 formed small plaques, while deletion of GAM-1 together with ORF22 resulted in even smaller ones in LMH cells. Simultaneous deletion of ORF28, ORF29, and GAM-1 led to growth defect of FAdV-4. These facts implied that genus-specific genes contributed to and synergistically affected viral replication, although no single one was essential. Notably, replication of FAdV-4 mutants could be different *in vitro* and *in vivo*. XGAM1-CX19A, a GAM-1-deleted FAdV-4 that replicated efficiently in LMH cells, did not kill chicken embryos because virus propagation took place at a very low level *in vivo*. This work laid a solid foundation for FAdV-4 vector construction as well as vaccine development, and would benefit viral gene function study.

**IMPORTANCE** Identification of viral essential genes is important for adenoviral vector construction. Deletion of nonessential genes enlarges cloning capacity, deletion of essential genes makes a replication-defective vector, and expression of essential genes in *trans* generates a virus packaging cell line. However, the genus-specific essential genes in FAdV have not been identified. We constructed adenoviral plasmid carrying deletions covering all 22 genus-specific ORFs of FAdV-4, and found that all virus mutants could be rescued and amplified in chicken LMH cells except those that had defects in key promoter activity. These genus-specific genes affected virus growth, but no single one was indispensable. Dysfunction of several genus-specific genes at the same time could make FAdV-4 vectors replication-defective. In addition, the growth of FAdV-4 mutants could be different in LMH cells and in chicken embryos, suggesting the possibility of constructing attenuated FAdV-4 vaccines.

**KEYWORDS** fowl adenovirus, essential gene, reverse genetics, virus rescue, RNA-seq, promoter, chicken embryo, reverse genetic analysis

Address correspondence to Zhuozhuang Lu, luzz@ivdc.chinacdc.cn, or Yingtao Lv, lvyingtao@qust.edu.cn.

The authors declare no conflict of interest.

Adenoviridae is classified into six genera, among which Mastadenoviruses infect mammalian hosts exclusively while Aviadenoviruses have been found only in birds (1). Mastadenoviruses, especially human adenovirus C (HAdV-C), have been intensively studied and constructed as gene transfer vectors (2–4). The application of HAdV-based vectors in human gene therapy and vaccine development has been hindered by high prevalence of pre-existing immunities against these pathogens in humans (5, 6), and interest has been attracted to construct recombinant fowl adenoviruses (FAdVs), a subgroup of aviadenoviruses, as gene delivery tools (7–9).

Fowl adenovirus contains a genome of linear double-stranded DNA of 43–46 kb in length, which is 8–10 kb longer than that of HAdV (10). Genus-common genes, encoding viral structural proteins and proteins involved in viral DNA replication and virion assembly, are located centrally in the genome. Genus- or species-specific genes, being involved in virus-host interaction, are mostly located near the ends of the genome (11). Genus-specific genes of HAdVs are further divided into E1, E3, and E4 regions with E1A gene being expressed firstly after virus infection (12–14). Most of HAdV genus-specific genes except those in the E3 region are conserved for all HAdV types, and their functions have been studied (11, 15, 16). In contrast, genus-specific genes of FAdV have no similarity to those of HAdV, and the functions of FAdV genus-specific genes have not been revealed, except for ORF1 (dUTPase), ORF8 (GAM-1) and ORF22 (17–19).

FAdV-4 is the predominant causative agent of hepatitis-hydropericardium syndrome (HHS) in chickens (20). Since July 2015, outbreaks of HHS caused by a novel genotype of FAdV-4 have been reported in China, causing severe economic losses to the poultry industry (21, 22). Previously, we constructed an infectious clone of the novel FAdV-4 and established a FAdV-4-based vector system (23–25). Here, we attempted to investigate the effects of 22 genus-specific genes on the virus replication by using reverse genetics approaches. Unexpectedly, we found no single genus-specific gene was essential for the replication of FAdV-4 in cell culture, although they could influence virus viability.

## RESULTS

**The strategy of constructing FAdV-4 mutants.** The restriction site-defined regions of the viral genome were systematically deleted to detect the key regions for virus replication. Frameshift mutations or coding sequence (CDS) deletions were further introduced into the key regions to distinguish the effects of individual genes in the following steps. To reduce the possible damage to mRNA splicing, deletion of a DNA fragment was generally carried out inside an ORF. In some cases, point mutations, which led to early termination of translation or frameshift mutations, were used to block the generation of functional gene products. The following procedure was utilized to generate FAdV-4 mutants: a DNA fragment was excised from the adenoviral plasmid by restriction digestion and used to construct an intermediate plasmid, in which more unique restriction sites could be used for site-directed modification; mutations were introduced into the intermediate plasmid; and finally, the modified intermediate plasmid was restored to the original adenoviral plasmid to generate a new one (26, 27). The start adenoviral plasmid pKFAV4-CX19A had the deletions of ORF1, ORF1B, and ORF2 at the left end and ORF19A at the right end of the FAdV-4 genome, and it also carried the insertion of CMV promoter (CMVp)-controlled mCherry expression cassette (24). The procedure of deleting HindIII-EcoRV fragment (XHE) at the right end of the genome is presented as an example (Fig. 1A). The generated intermediate plasmid pAMS9002 and adenoviral plasmid pKXHE-CX19A were identified by restriction analysis (Fig. 1B and C). Recombinant virus XHE-CX19A was rescued from PmeI-linearized pKXHE-CX19A-transfected Leghorn male hepatoma (LMH) cells. XHE-CX19A was identified by restriction analysis of the viral genome, PCR amplification of the HindIII-EcoRV fusion site, and sequencing (Fig. 1D to F). Other FAdV-4 mutants were similarly constructed.

**Rescue and growth of FAdV-4 viruses with restriction sites-defined deletions at the right end.** As shown in Fig. 2A, the right end of the FAdV-4 genome could be divided into 6 fragments by 7 restriction sites. Adenoviral plasmids carrying deletions of

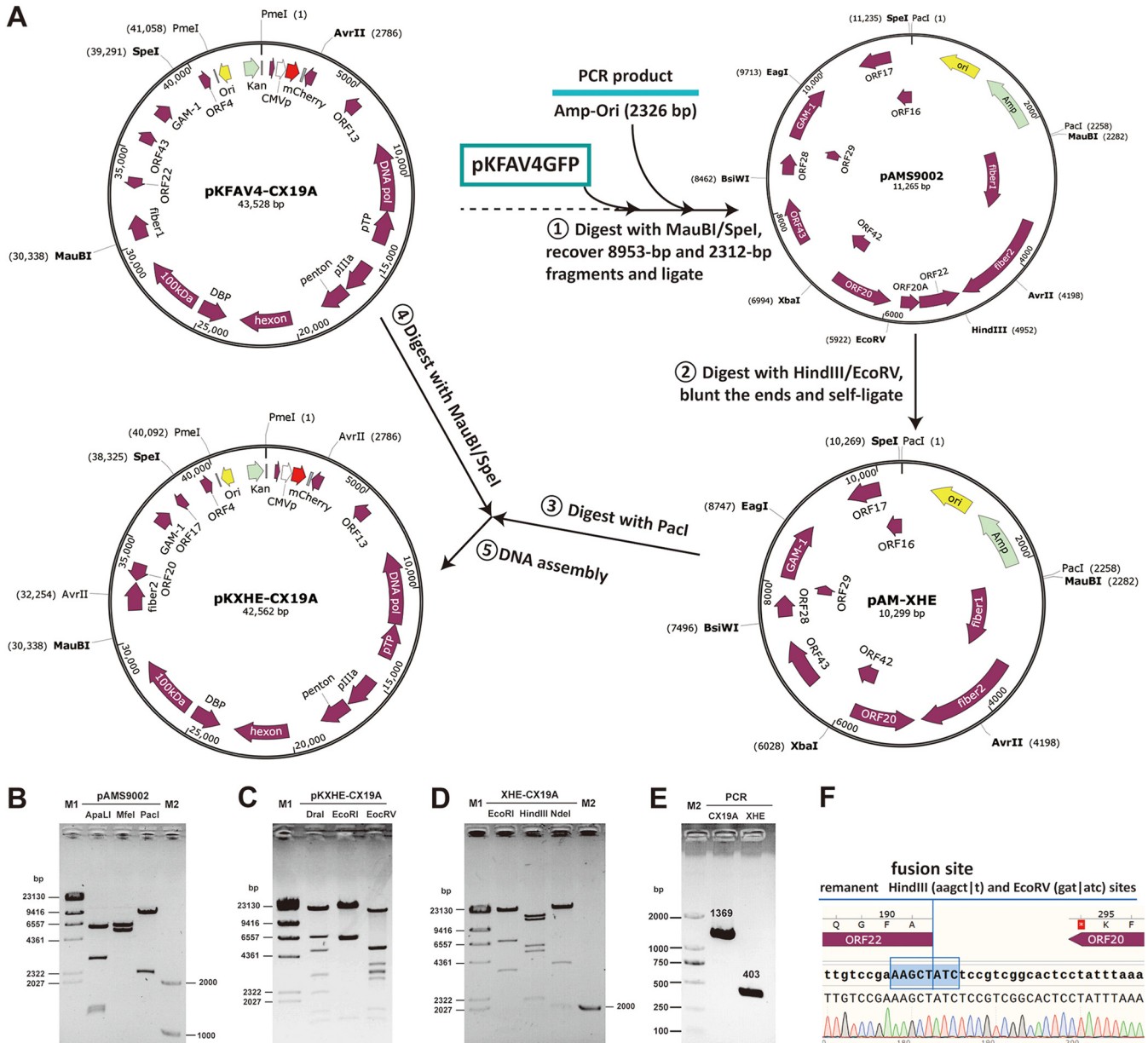

**FIG 1** Construction and identification of recombinant FAdV-4 virus carrying the deletion of the HindIII-EcoRV fragment at the right end of the genome (XHE-CX19A). (A) Schematic diagram of constructing adenoviral plasmid pKXHE-CX19A. Combined use of pKFAV4-CX19A and the intermediate plasmid pAMS9002 generated a series of FAdV-4 mutants carrying deletions from the EcoRV site to the SpeI site at the right end of the genome. Construction of pKXHE-CX19A is shown as an example here to demonstrate the procedure of genetic modification. (B) Restriction analysis of the intermediate plasmid pAMS9002. The predicted molecular weights (bp) of digested fragments were 1246, 1312, 2872, and 5835 for ApaLI; 5240 and 6025 for MfeI; and 2257 and 9008 for PacI. (C) Restriction analysis of the adenoviral plasmid pKXHE-CX19A. The predicted molecular weights (bp) of digested fragments were 19, 698, 839, 1360, 1613, 2209, 3001, 5000, 6947, and 20876 for DraI; 1384, 6545, 6859, and 27774 for EcoRI; and 176, 645, 685, 1365, 1595, 2643, 2974, 3038, 3519, 4988, 5016. and 15918 for EcoRV. (D) Restriction analysis of XHE-CX19A genomic DNA. The predicted molecular weights (bp) of digested fragments were 516, 1384, 3552, 6859, and 27774 for EcoRI; 2119, 5216, 5900, 12356, and 14494 for HindIII; and 251, 1016, 1906, 3961, and 32951 for NdeI. (E) Identification of the HindIII-EcoRV deletion in the XHE-CX19A genome by PCR. PCR was performed to amplify the fragment spanning the deletion region by using genomic DNA of FAdV4-CX19A (CX19A) or XHE-CX19A (XHE) as the templates, respectively. The products were 1369 or 403 bp in length. (F) Identification of the HindIII-EcoRV deletion in the XHE-CX19A genome by sequencing the PCR product of 403 bp.

the 6 fragments were constructed. The other two plasmids, pKXHB-CX19A and pKXXS-CX19A, carried deletions spanning 3 restriction fragments. PmeI-linearized adenoviral plasmids were transfected into chicken hepatoma LMH cells (28). Few small fluorescence foci could be observed on pKXBE-CX19A, pKXES-CX19A, or pKXXS-CX19A-transfected cells; and these small foci did not grow or even disappeared as the culture time prolonged. For the remaining 5 plasmids, mCherry foci could be observed 2 or 3 days

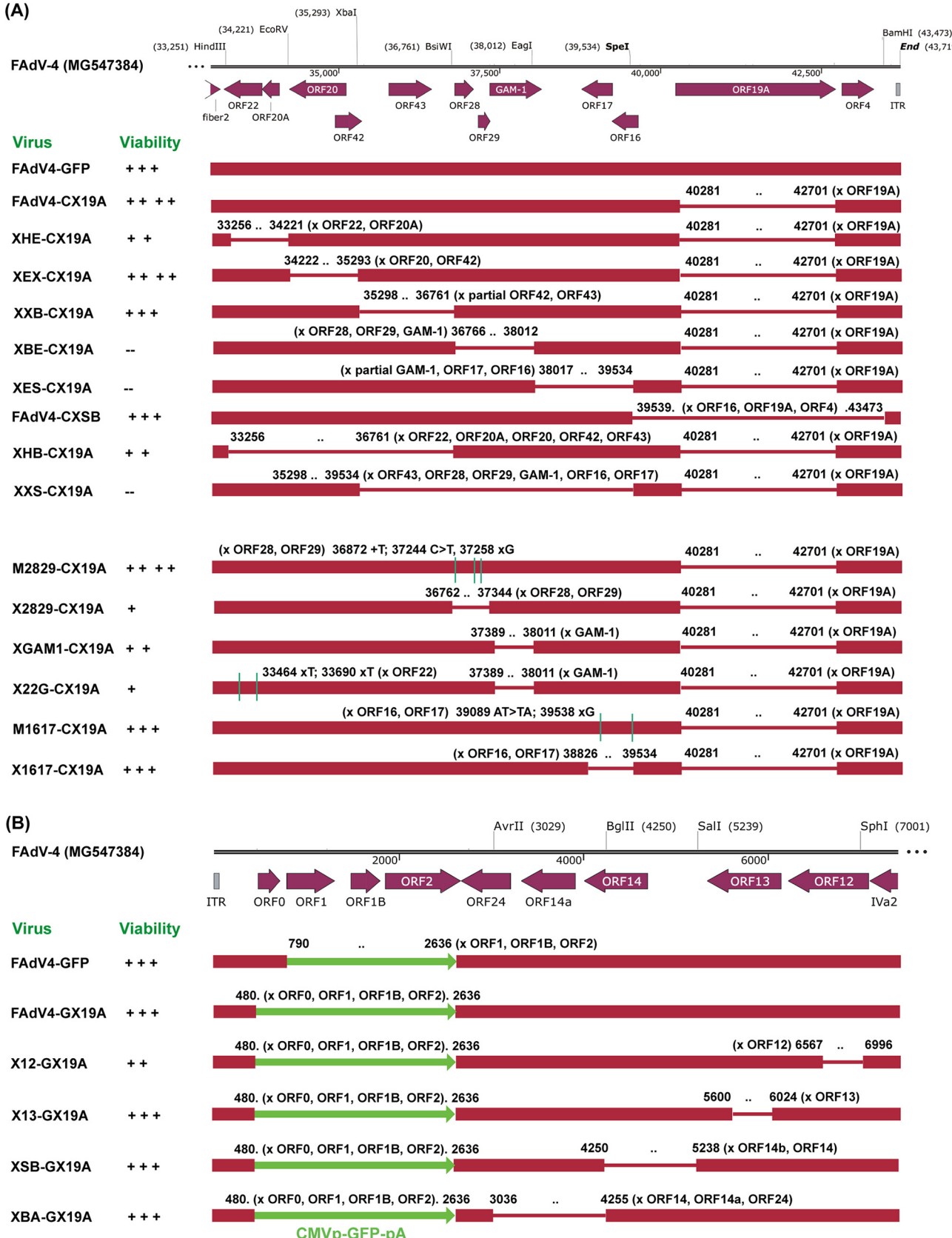

**FIG 2** Schematic diagram of FAdV-4 mutants. The viability of recombinant viruses was evaluated according to the plaque size data. The viability of FAdV4-GFP was set as the standard for comparison and labeled with "+++". The boundaries of deletions or point mutations were labeled with

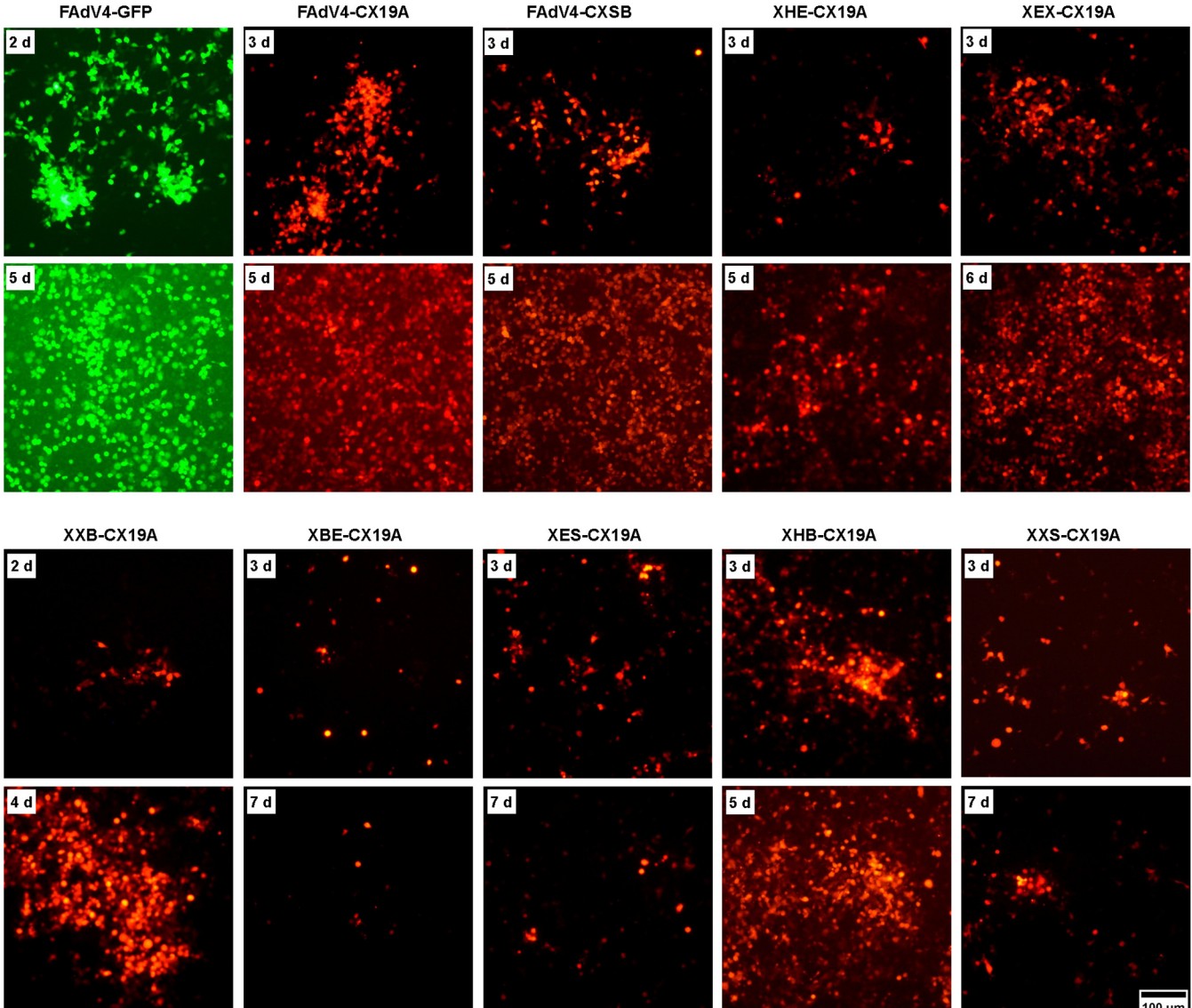

**FIG 3** Rescue of FAdV-4 mutants carrying deletions at the right end of the viral genome in LMH cells. PmeI-linearized adenoviral plasmids were used to transfect LMH cells, and the expression of the reporter gene was observed under fluorescence microscope in the following days. FAdV4-GFP served as the control. The occurrence and growth of fluorescence foci implied a successful rescue of recombinant viruses.

posttransfection, they kept growing, and complete cytopathic effect (CPE) occurred as the borders of foci merged (Fig. 3). Previously constructed pKFAV4-GFP and pKFAV4-CX19A served as positive controls (24).

Plaque forming experiments were carried out to compare the viability of rescued FAdV-4 mutants. The morphology of plaques formed by some FAdV-4 mutants is shown in Fig. 4A. Plaque sizes were then measured and normalized (Fig. 4B). Compared to the control FAdV4-GFP, deletion of EcoRV-XbaI fragment (XEX) enhanced viral growth, and deletion of HindIII-EcoRV (XHE), XbaI-BsiWI (XXB), or SpeI-BamHI (XSB) fragments did not obviously affect viral growth. Interestingly, the virus could grow even if 3 adjacent fragments from HindIII to BsiWI sites were simultaneously

**FIG 2** Legend (Continued)
numbers, which corresponded to the nucleotide (nt) sites in the wild-type FAdV-4 genome (GenBank accession number MG547384). (A) Viruses carrying deletions at the right end. Each of these FAdV-4 mutants also contained the deletion of ORF1-ORF1B-ORF2 and the insertion of the reporter gene at the left end of the genome. (B) Viruses carrying deletions at the left end. Each of these FAdV-4 mutants also contained the deletion of ORF19A at the right end of the genome, except FAdV4-GFP.

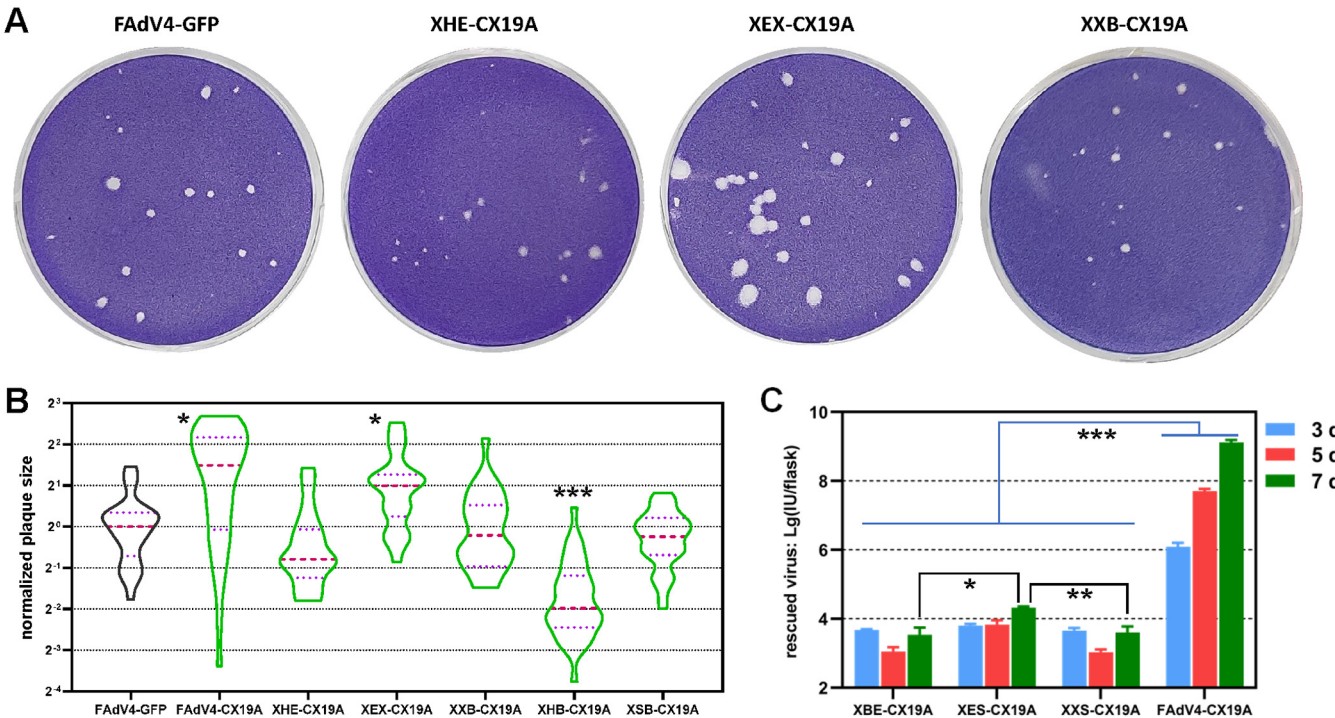

**FIG 4** Growth of FAdV-4 mutants carrying deletions at the right end of the viral genome in LMH cells. (A) Recombinant viruses formed plaques on the LMH monolayer in semi-solid cell culture system 7 days postinfection (dpi). (B) Violin plots of the plaque size data. The areas of 39, 32, 39, 30, 36, and 37 plaques were measures for FAdV4-CX19A, XHE-CX19A, XEX-CX19A, XXB-CX19A, XHB-CX19A, and XSB-CX19A viruses, respectively. About 30 plaques were included for the control FAdV4-GFP virus for each batch of experiments. The data collected were normalized to the median size of plaques formed by FAdV4-GFP in the same batch of experiments, and the median sizes of plaques formed by recombinant viruses were compared to that formed by the control FAdV4-GFP by using the nonparametric Mann-Whitney test. (C) Titration of rescued XBE-, XES-, and XXS-CX19A. FAdV4-CX19A served as a control. Viruses were collected from adenoviral plasmid-transfected LMH cells 3, 5, or 7 days posttransfection. The data were logarithmically transformed and statistically analyzed by using two-way analysis of variance (ANOVA). The multiple comparisons were carried out between virus mutants at day 7 posttransfection. (*, $P < 0.05$; **, $P < 0.01$; ***, $P < 0.001$).

deleted (XHB), but this mutant formed the smallest plaques. Few viruses could be rescued, and they could hardly grow to form plaques if either fragment between BsiWI/EagI/SpeI (XBE, XES, or XXS) was deleted.

To describe the viability of XBE-CX19A, XES-CX19A, and XXS-CX19A more accurately, detached cells were collected 3 or 5 days post plasmid transfection, and cells together with culture medium were harvested 7 days posttransfection. The associated viruses were titrated. For XBE-CX19A and XXS-CX19A, the rescued viruses at day 3 were in a higher amount than those at day 5 or even day 7. In contrast, XES-CX19A could grow as the culture time prolonged. However, the growth of XES-CX19A was slow and inefficient. The overall yield of XES-CX19A at day 7 posttransfection was 4 orders of magnitude lower than that of FAdV4-CX19A (Fig. 4C). These results demonstrated that progeny viruses could be rescued from pKXBE-CX19A, pKXES-CX19A, or pKXXS-CX19A-transfected LMH cells, but the virus yields were very low and the growth of progeny viruses could not be sustained.

**Virus rescue for adenoviral plasmids carrying deletions at the left end of the genome.** The start adenoviral plasmid pKFAV4-CX19A carried deletions in ORF1, ORF1B, and ORF2 at the left end of the genome. ORF0 was further deleted and mCherry CDS was replaced with that of GFP in pKFAV4-CX19A to generate pKFAV4-GX19A. Based on pKFAV4-GX19A, other ORFs at the left end were deleted sequentially to construct adenoviral plasmids. Viruses carrying these deletions were all rescued from transfected LMH cells (Fig. 2 and 5). It seemed that X12-GX19A and X13-GX19A produced smaller plaques than FAdV4-GFP did. However, only the difference between X12-GX19A and FAdV4-GFP was statistically significant ($P < 0.05$). FAdV4-GX19A and XSB-GX19A formed plaques in a similar size to those formed by FAdV4-GFP, while those produced by XBA-GX19A appeared to be larger, though the difference was not

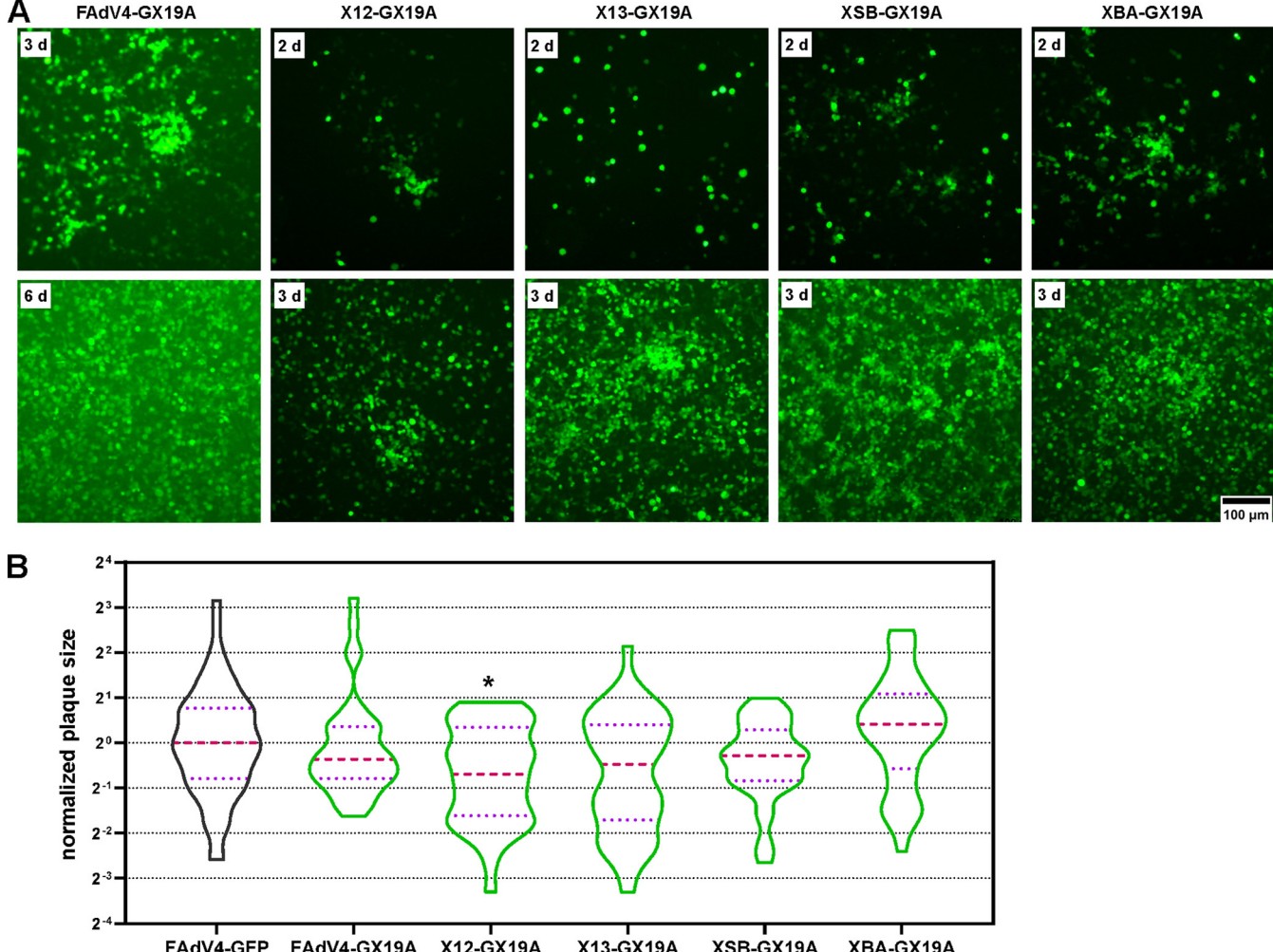

**FIG 5** Rescue and growth of FAdV-4 mutants carrying deletions at the left end of the genome. (A) PmeI-linearized adenoviral plasmids were used to transfect LMH cells, and the expression of GFP was observed under a fluorescence microscope. The occurrence and growth of fluorescence foci suggested a successful rescue of recombinant virus. (B) Violin plots of the plaque size data. The areas of 37, 43, 39, 37, and 44 plaques were measures for FAdV4-GX19A, X12-GX19A, X13-GX19A, XSB-GX19A, and XBA-GX19A viruses, respectively. About 30 plaques were included for the control FAdV4-GFP virus for each batch of experiments. The data collected were normalized to the median size of plaques formed by FAdV4-GFP in the same batch of experiments, and the median sizes of plaques formed by recombinant viruses were compared to that formed by the control FAdV4-GFP by using the nonparametric Mann-Whitney test. (*, $P < 0.05$).

significant (Fig. 5). These results indicated that genus-specific genes at the left end were dispensable for the replication of FAdV-4.

**Precise inactivation of ORF28, ORF29, GAM-1, ORF16, or ORF17 in BsiWI-SpeI region.** Systematic deletion experiments indicated that the BsiWI-SpeI region at the right end played a key role in virus rescue and replication. Five ORFs were annotated in this region. Frameshift mutations or deletions were introduced into these ORFs to inactivate functional expression. Frameshift mutations in ORF28 and ORF29 (M2829-CX19A) had no influence on the virus growth, while deletion of them (X2829-CX19A) severely impaired virus replication (Fig. 6A and B). Transfection experiments showed that GAM-1-deleted XGAM1-CX19A could be rescued and the plaques it formed were smaller in size than those formed by FAdV4-GFP. When the expression of ORF22 was further blocked by introducing frameshift mutations, X22G-CX19A was also successfully rescued, and the plaques formed by X22G-CX19A were smaller than those formed by XGAM1-CX19A (Fig. 6A and B). Considering that XHE-CX19A was viable, in which ORF22 was completely deleted (Fig. 3A and B), these results demonstrated that GAM-1 and ORF22 contributed synergistically to the growth of FAdV-4, although neither was an essential gene. Frameshift and deletion in ORF16 and ORF17 had little influence on

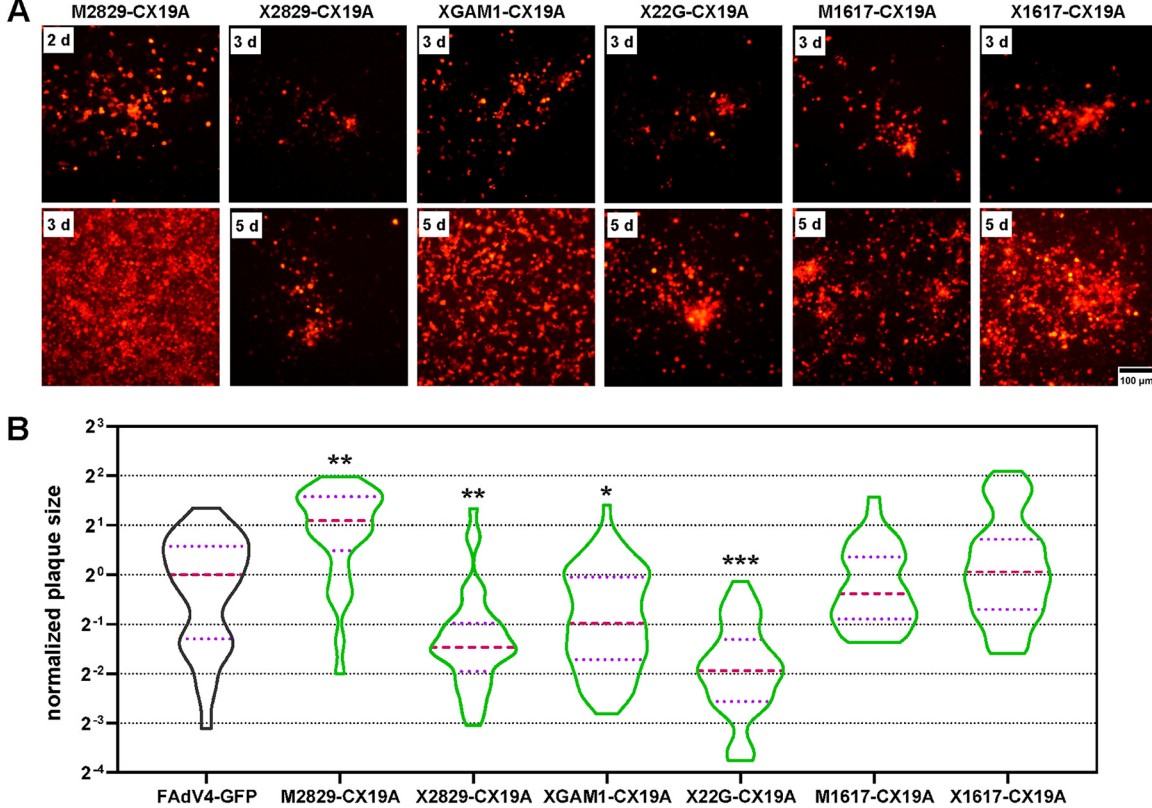

**FIG 6** Rescue and growth of recombinant FAdV-4 viruses carrying deletions in the BsiWI-SpeI region at the right end of the genome. (A) PmeI-linearized adenoviral plasmids were used to transfect LMH cells, and the expression of mCherry was observed and photographed under fluorescence microscope in the following days. The occurrence and growth of fluorescence foci suggested a successful rescue of recombinant virus. (B) Violin plots of the plaque size data. The sizes of 30, 41, 68, 37, 39, and 37 plaques were measures for M2829-CX19A, X2829-CX19A, XGAM1-CX19A, X22G-CX19A, M1617-CX19A, and X1617-CX19A viruses, respectively. About 30 plaques were included for the control FAdV4-GFP virus for each batch of experiments. The data collected were normalized to the median size of plaques formed by FAdV4-GFP in the same batch of experiments, and the median sizes of plaques formed by recombinant viruses were compared to that formed by the control FAdV4-GFP by using the nonparametric Mann-Whitney test (\*, $P < 0.05$; \*\*, $P < 0.01$; \*\*\*, $P < 0.001$).

the growth of recombinant viruses. These data suggested that ORF28, ORF29. and GAM-1 played important roles in virus replication, and the crucial functions of ORF28 and ORF29 did not result from their protein encoding ability.

**Plaque forming ability of FAdV-4 mutants in primary chicken cells.** The growth of virus in primary cells could be different from that in the hepatoma cell line due to altered cell signaling in cancer cells. Plaque forming experiments were conducted on primary chicken embryo hepatocytes for the 18 rescued FAdV-4 mutants (Fig. 7). All these viruses grew and formed plaques on monolayers of primary hepatocytes. Compared to FAdV4-GFP, some mutants propagated differently in normal and malignant cells. XEX-CX19A and M2829-CX19A produced larger plaques in LMH cells, and X12-GX19A produced smaller ones in LMH, while they formed plaques in a normal size in primary cells. XHE-CX19A, FAdV4-GX19A, and XSB-GX19A formed larger plaques than FAdV4-GFP in primary cells, and FAdV4-CXSB, X1617-CX19A, and X13-GX19A grew smaller ones in primary cells, while these mutants formed plaques in a regular size in LMH cells. The plaque forming ability of others did not change significantly between LMH cells and primary hepatocytes. For example, FAdV4-CX19A formed larger plaques than FAdV4-GFP on both cell monolayers, while XHB-CX19A, X2829-CX19A, XGAM1-CX19A, and X22G-CX19A produced smaller ones. It is worth noting that no inversive change of virus viability was observed. Namely, there was no such mutant that formed larger plaques than FAdV4-GFP in LMH cells but produced smaller ones in primary hepatocytes, or vice versa.

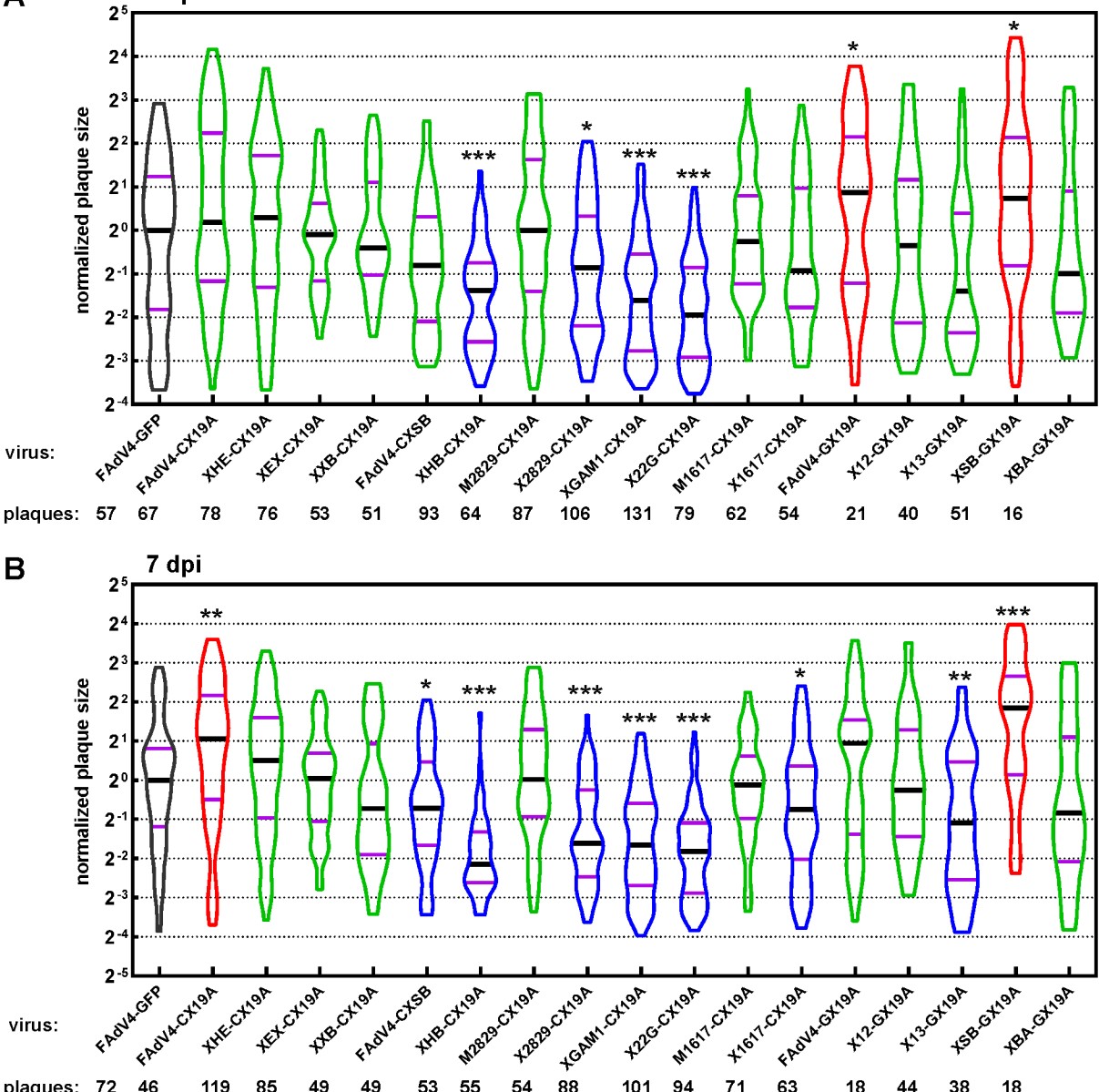

**FIG 7** Growth of FAdV-4 mutants in primary chicken embryo hepatocytes. Chicken embryo hepatocytes were isolated and infected with FAdV-4 recombinant viruses. The GFP or mCherry foci were photographed after 6 or 7 days' cultivation in semi-solid media. The areas of these foci (plaques) were measured. The data collected were normalized to the median size of plaques formed by FAdV4-GFP, and the median sizes of plaques formed by recombinant viruses were compared to that formed by the control FAdV4-GFP by using the nonparametric Mann-Whitney test. Shown are the violin plots of the plaque size data collected 6 days postinfection (dpi) (A) and 7 dpi (B). The numbers of plaques photographed were annotated below the names of virus mutants. (*, $P < 0.05$; **, $P < 0.01$; ***, $P < 0.001$).

**The complementary effect of GAM-1 helper plasmid.** GAM-1 is an important gene that has been identified as a functional homolog to human adenovirus E1B19K protein and could prevent the infected cells from apoptosis in the early phase of the virus life cycle (29, 30). The N-terminal 221 amino acid (aa) residues of GAM-1 were in the BsiWI-EagI fragment, and the other 50 aa at the C-terminal were in the EagI-SpeI fragment. In XBE-CX19A and XXS-CX19A, the expression of GAM-1 was totally excluded due to sequence deletion or sequence deletion together with frameshift. In XES-CX19A, GAM-1 might be expressed as a truncated protein with a normal N-terminal of 222 aa and a short frame-shifted C-terminal. Lack of GAM-1 might be responsible for the defective growth of XBE-CX19A, XES-CX19A, and XXS-CX19A. CMVp-controlled GAM-1 expression plasmid (pcDNA3-GAM1) was

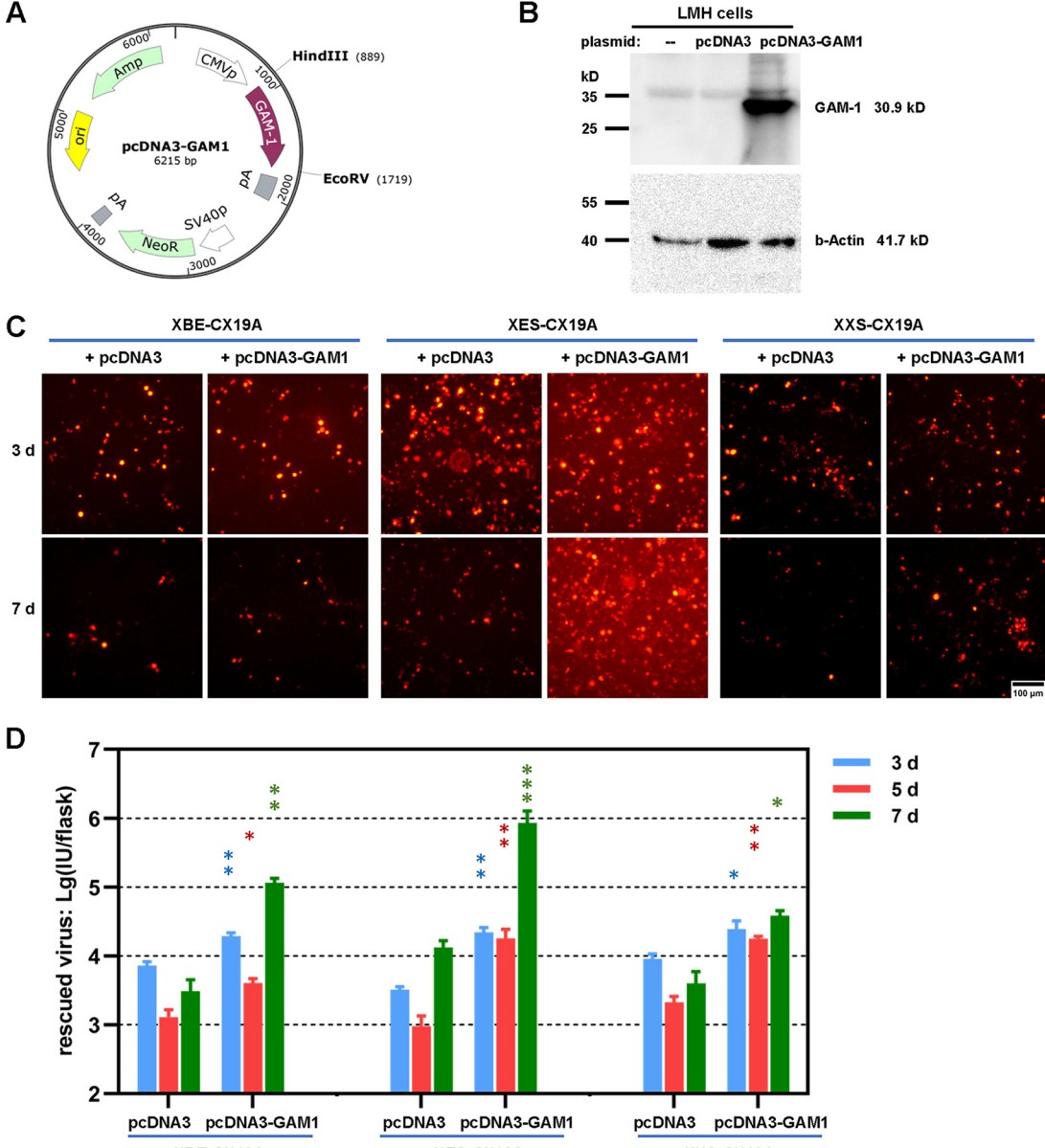

**FIG 8** The effect of the helper plasmid carrying the viral GAM-1 gene on the rescue of FAdV-4 mutants with deletions in the BsiWI-SpeI region. (A) The map of the eukaryotic expression plasmid carrying the GAM-1 gene (pcDNA3-GAM1). The expression of GAM-1 was controlled by CMV promoter (CMVp) in this plasmid. (B) Detection of GAM-1 expression in pcDNA3-GAM1-transfected LMH cells by Western blotting. Proteins extracted from untransfected or pcDNA3-transfected LMH cells were loaded as negative controls. Rabbit anti-GAM1 sera were prepared by immunizing animals with purified 6×His tagged prokaryotically expressed GAM-1 protein and used as the primary antibody. $\beta$-actin was stained as a sample-loading control. (C) Transfection of LMH cells with the mixture of adenoviral plasmid and helper pcDNA3-GAM1. The expression of mCherry reporter was observed under fluorescence microscope in the following days. (D) Titration of the rescued viruses collected from transfected LMH cells 3, 5, or 7 days posttransfection. The data were logarithmically transformed and statistically analyzed by using a two-way ANOVA. The multiple comparisons were carried out between pcDNA3 and pcDNA3-GAM1 groups within each time point for each viral mutant (*, $P < 0.05$; **, $P < 0.01$; ***, $P < 0.001$).

constructed (Fig. 8A), and the expression of GAM-1 was verified in pcDNA3-GAM1 transfected LMH cells by Western blotting (Fig. 8B). A mixture of pcDNA3-GAM1 and linearized adenoviral plasmids was used to transfect LMH cells. Without pcDNA3-GAM1, the amount of mCherry+ cells gradually reduced as the culture time was prolonged. When pcDNA3-GAM1 was included, this trend slowed down. XES-CX19A was different from XBE-CX19A and XXS-CX19A in that more mCherry+ cells and even small fluorescence foci could be observed 7 days post the transfection of the mixture

of pKXES-CX19A and pcDNA3-GAM1 (Fig. 8C). Rescued viruses in detached cells were titrated, and it was observed that addition of pcDNA3-GAM1 significantly increased the yields of rescued viruses (Fig. 8D). For XES-CX19A, the virus yield increased nearly 2 orders of magnitude at day 7 when pcDNA3-GAM1 was included. Considering the limited plasmid transfection efficiency in LMH cells, only a part of the cells expressed GAM-1 protein. It was understandable that cotransfection with pcDNA3-GAM1 could hardly help XES-CX19A form foci of regular size and thus sustain virus growth. However, it was clearly seen that expression of GAM-1 partially complemented the deletions of viral genome in the BsiWI-SpeI region, especially in the EagI-SpeI region.

**Promoter activities in BsiWI-EagI and EagI-SpeI fragments.** Since blocking the expression of ORF28, ORF29, GAM-1, ORF16, or ORF17 could not completely explain the defective growth of XBE-CX19A and XES-CX19A, there might be cis-acting elements, such as promoters, located in these regions. The sequence from the 3′ end of ORF43 to the 5′ end of GAM-1 was cloned and named RP1 (right-end promoter 1); mCherry and GFP were added to the 5′ and 3′ ends of RP1, respectively, to form a plasmid, and we designated this RP1-reporters construct mCherry-RP1-GFP. Similarly, the sequence from the 3′ end of GAM-1 to the 5′ end of ORF19A (RP2) was cloned, GFP and mCherry were added to the 5′ and 3′ ends of RP2, respectively, and this construct was called GFP-RP2-mCherry (Fig. 9A). Plasmids carrying CMVp-controlled GFP or mCherry served as positive controls. These plasmids were transfected into LMH cells, and the expression of GFP and mCherry was observed under a fluorescence microscope (Fig. 9B). Apparently, the transcription was more active rightward for both RP1 and RP2, especially for RP1. The data of flow cytometry assay verified this finding (Fig. 9C). We assumed that the transfection efficiency was the same for all the 4 plasmids, and therefore the geometric mean of fluorescence intensity for all detected cells could represent the activity of promoters (Fig. 9D). The leftward or rightward activities of RP1 and RP2 relative to that of CMVp are shown in Fig. 9E. RP1 had considerable rightward promoter activity, which was approximately half that of CMVp. The rightward activity of RP2 was 7% of that of CMVp. The leftward promoter activity was detectable, but very weak. These data illustrated that RP1 and RP2, especially RP1, had remarkable rightward promoter activity in chicken LMH cells.

**Promoter and introns inferred from RNA-seq data.** RNA-seq experiments were performed to study the transcription of the virus genome. Reads alignments to the virus genome from ORF43 to GAM-1 are shown in Fig. 10A. ORF43 and GAM-1 genes were normally transcribed. In contrast, the transcription of ORF28 and ORF29 was only detected at the very ends of the ORFs. It could be clearly observed from the splice junction tracks 12 h postinfection (hpi) that two introns spanned the ORF29 region. This pattern of transcription also implied that a promoter (RP1) was located at ORF28 considering that neither transcription nor rightward mRNA splicing were seen upstream of ORF29. After zooming in, the reads coverage at the 3′ ends of ORF28 and ORF29 could be observed at single-nucleotide resolution. The transcription start site (TSS) could be the nucleotide A in sequence ggttcagacA located at the 3′ end of ORF28 (position 37061 bp in the FAdV-4 genome), and the borders of the intron around ORF29 were unambiguously defined (Fig. 10A).

The promoter around ORF28 (RP1) was further analyzed by using bioinformatics software. The TATA box and the initiator could be identified with high confidence by many core promoter prediction programs. Because the information of chicken transcription factor binding site (TFBS) was scarce, the sequence was scanned with JASPAR CORE matrices of vertebrate TFBS and annotated (Fig. 10B). Many TFBSs were discovered in this region. Interestingly, 7 HNF1A binding sites were located upstream of the predicted TATA box. HNF1A (hepatocyte nuclear factor 1-alpha) activates the tissue specific expression of multiple genes, especially in the liver, pancreas, and intestine (31).

**The growth of FAdV-4 mutants in chicken embryos.** The growth of FAdV-4 mutants in chicken embryos might be different from that in LMH cells. XHE-CX19A, which had a deletion of ORF22, and XGAM1-CX19A, which had a deletion of GAM-1, were chosen to inoculate 6-day-old chicken embryos. FAdV4-GFP and phosphate-buffered saline (PBS) served as positive and negative controls, respectively. As shown

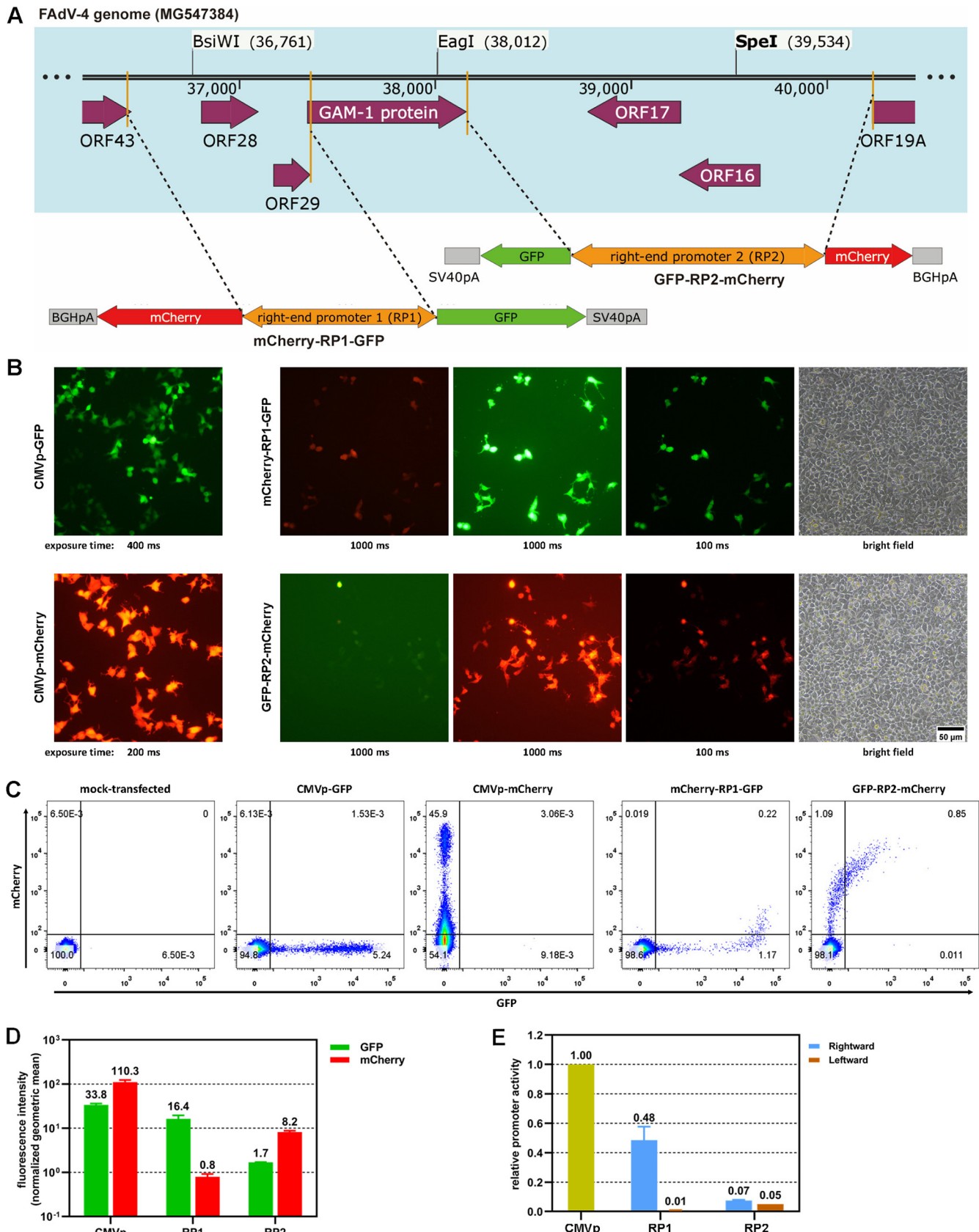

**FIG 9** Promoter activity around the BsiWI-SpeI region at the right end of the FAdV-4 genome. (A) schematic diagram of constructing reporter plasmids to evaluate promoter activity. The cloned FAdV-4 fragments were designated RP1 (right-end promoter 1) and RP2, respectively. (B) Expression of reporter

in Fig. 11A, embryos in the FAdV4-GFP group all died, 4 of 14 embryos died in XHE-CX19A group, and all embryos survived in the XGAM1-CX19A and PBS groups. Viruses from livers of embryos were titrated on LMH cells. In the XHE-CX19A group, the virus yield in the livers of dead embryos was not statistically different from that of living ones. The mean virus yield in the FAdV4-GFP group was about 7 times higher than that in the XHE-CX19A group, while the virus yield in the XGAM1-CX19A group was approximately 4 orders of magnitude lower than that in XHE-CX19A group (Fig. 11B). Notably, XHE-CX19A and XGAM1-CX19A had similar viability when evaluated in LMH cells, while XGAM1-CX19A formed smaller plaques than XHE-CX19A in primary chicken hepatocytes. These data indicated that deletion of ORF22 slightly reduced the viral virulence while deletion of GAM-1 significantly decreased FAdV-4 replication in chicken embryos.

## DISCUSSION

No genus-specific gene was found essential for the replication of FAdV-4 in LMH cells. Genus-common genes are in the central genome, from IVa2 to fiber2, and the genus-specific genes are at the left or right ends (11, 17, 32). It was reported that ORFs 0, 1, 1A, 1B, 1C, and 2 at the left and TR-2, ORFs 11, 17, and 19 at the right of the genome were nonessential genes for FAdV-9, and ORFs 16, 17, and 19 were dispensable for FAdV-4 (8, 18). Point mutations were introduced to generate FAdV-A1 (CELO virus) mutants, and it was found that 16 of the 22 ORFs were nonessential (33). These studies did not include all genus-specific genes and were not comprehensive. Besides, there is considerable difference between homologous genes in various FAdV types, and FAdVs even contain some species-specific genes (17, 34). The effect of genus-specific genes on FAdV-4 replication has not been systematically studied. We constructed adenoviral plasmids carrying a series of restriction sites-defined deletions at both ends of the FAdV-4 genome. All virus mutants except those carrying deletions in the BsiWI-EagI-SpeI region could be rescued from plasmids-transfected packaging cells, and they could form plaques in LMH cells (Fig. 2A). The ORFs in the BsiWI-EagI-SpeI region, including ORF28, ORF29, GAM-1, ORF16, and ORF17, were further investigated. Deletions of GAM-1 or ORF16-ORF17 did not severely influence the replication of recombinant viruses. Frameshift mutations were introduced to interrupt the translation of ORF28 and ORF29, and the generated virus mutant grew as efficiently as the parental virus (FAdV4-CX19A), suggesting that the possible protein products of ORF28 and ORF29 were also dispensable. All FAdV-4 mutants, which were successfully rescued from LMH cells, grew and formed plaques in primary chicken hepatocytes (Fig. 7), suggesting the conclusion about there were no essential genus-specific genes could be extended to cultured normal chicken cells.

Promoter activity was discovered in the ORF28 region. It was predicted by a computational tool that there were promoters located in the BsiWI-SpeI region, especially in ORF28, and ORF29 was an upstream ORF (uORF, located upstream of the real CDS to reduce the translation of a gene's transcript) with low coding potential (17). The promoter hypothesis was testified by reporter genes transfection experiments, and the results showed that the region between ORF43 and GAM-1 (RP1) had considerable rightward promoter activity (Fig. 9). The RNA-seq data also suggested a promoter at ORF28. In addition, the data also revealed an intron at ORF29 and another one spanning ORF29 and GAM-1 (Fig. 10). Taken together, ORF28 and ORF29 were not real virus genes, a promoter was located at ORF28, and this promoter (RP1) controlled the expression of GAM-1 and even other genes downstream of GAM-1, which explained

**FIG 9** Legend (Continued)
genes observed under a fluorescence microscope. Reporter plasmids were transfected into LMH cells and the expression of GFP or mCherry was observed 48 h posttransfection. Plasmids carrying CMV promoter (CMVp)-controlled GFP or mCherry served as positive controls (CMVp-GFP and CMVp-mCherry). The exposure times were provided under each photograph. (C) Expression of reporter genes determined by flow cytometer. The expression of GFP and mCherry was demonstrated in pseudocolor dot plots. (D) Geometric mean fluorescence intensity of total cells. The data were normalized by subtracting the background value of mock-transfected samples from the value of CMVp controls and test groups. (E) Relative promoter activity to that of CMVp. The normalized geometric mean GFP or mCherry intensities of samples in test groups were divided by those of CMVp-GFP or CMVp-mCherry, respectively. The resulting values represented relative promoter activity. It could be seen that RP1 and RP2, especially RP1, had considerable rightward promoter activity.

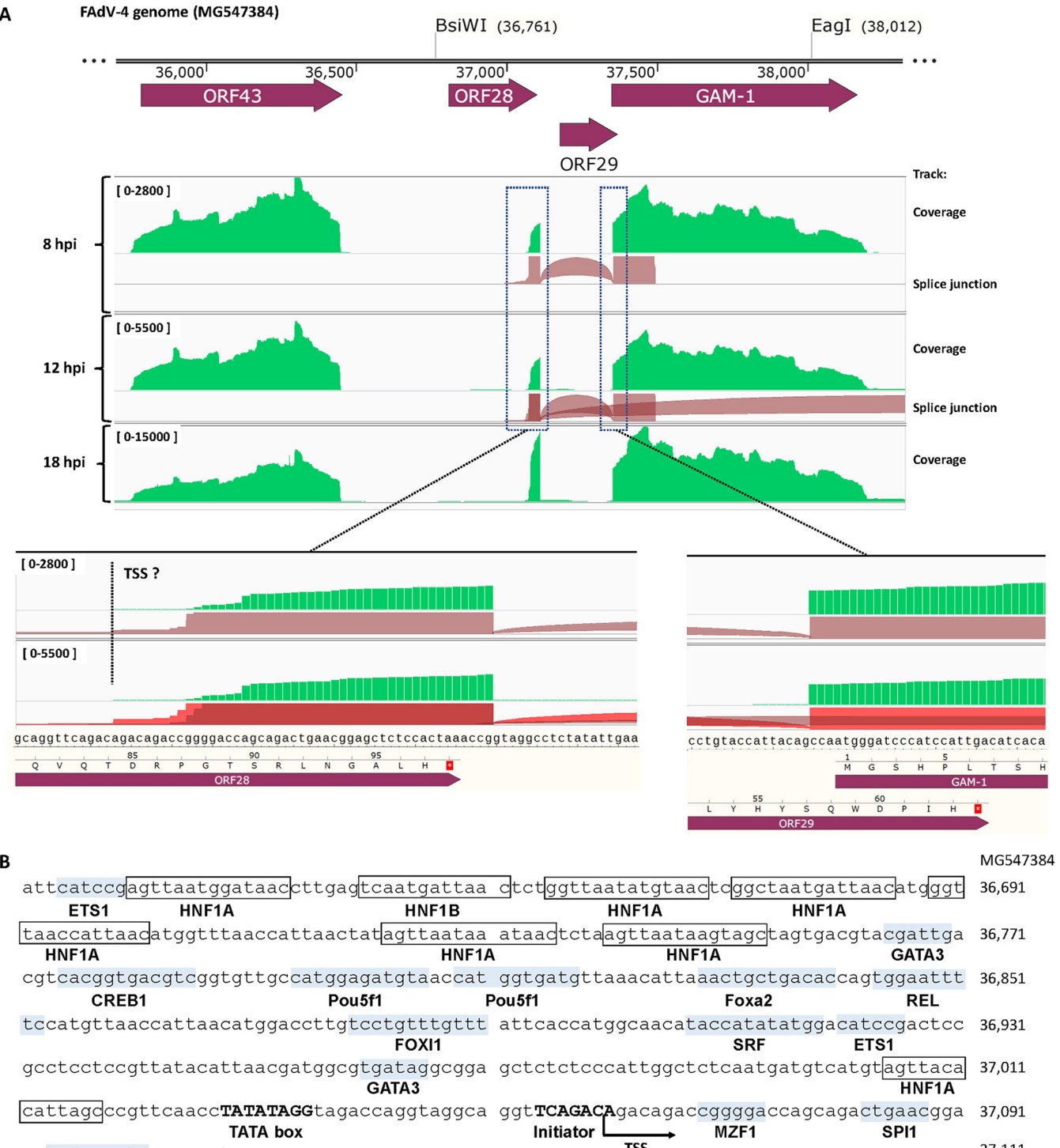

**FIG 10** Promoter and introns in the BsiWI-EagI region of the FAdV-4 genome inferred from RNA-seq data. (A) Genome-browser view of RNA-seq signals around the BsiWI-EagI region. RNA was extracted from FAdV4-GFP- or FAdV4-CX19A-infected chicken LMH cells at 8, 12, and 18 h postinfection (hpi). After poly(A) mRNA enrichment, samples were subjected to an RNA-seq pipeline for nonstranded 150 bp paired-end sequencing. HISAT2 was used to map RNA-seq reads to the virus genomes, and the Integrative Genomics Viewer (IGV) program was used to view the sorted alignments. IGV screenshots were edited to show the reads alignments of FAdV4-CX19A RNA-seq data around the BsiWI-EagI region. It could be deduced that a promoter was in ORF28 while two introns spanned ORF29. The FAdV4-GFP data gave the same results (not shown). (B) *In-silico* annotation of the predicted promoter located in ORF28. The core promoter elements, including TATA box and initiator in ORF28, were predicted with the Neural Network Promoter Prediction (NNPP) and ElemeNT programs. Sequence ranging from −450 to +50 nucleotides (nt) relative to the predicted +1 nt transcription start site (TSS) was further analyzed with the LASAGNA-Search program to find potential transcription factor binding sites (TFBSs). Because chicken TFBS information is very limited, JASPAR CORE matrices of vertebrates were selected as the transcription factor (TF) model input in the program. The predicted TFBSs and the corresponding transcription factors were further confirmed by using the tools of transcription factor DNA-binding matrix searching and scanning on the JASPAR website.

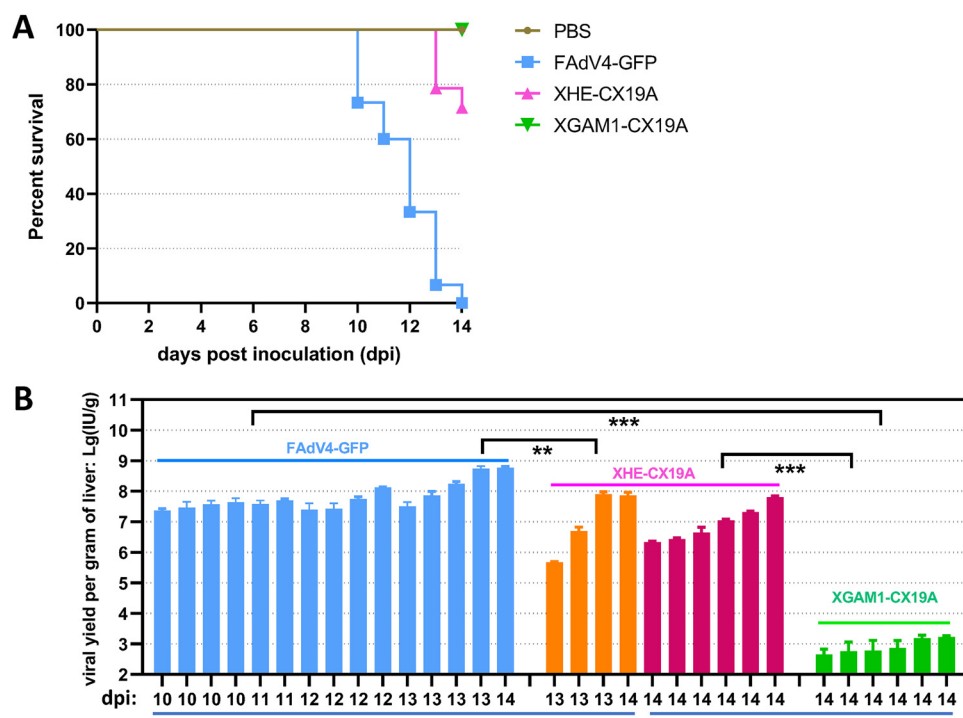

**FIG 11** Growth of recombinant FAdV-4 viruses in chicken embryos. Six-day-old embryonated chicken eggs were inoculated with FAdV4-GFP, XHE-CX19A, or XGAM1-CX19A of $1 \times 10^8$ vp in 100 μL PBS via the yolk sac route, respectively (15 eggs per group). The PBS group was injected with 100 μL PBS containing no virus, and served as mock-infected control (15 eggs). The viability of embryos was checked every 24 h. The viable embryos were killed by chilling the eggs at 4°C overnight 14 days postinoculation (dpi). Livers were collected and weighed for virus titration. (A) Survival curve of embryonated chicken eggs after viral inoculation. (B) Virus yield in liver normalized by liver weight, which was shown as infectious units per gram of liver (IU/g). Livers from dead embryos were all included for virus titration, and six livers were randomly selected and titrated for viable embryos from each test group of XHE-CX19A and XGAM1-CX19A.

why frameshift mutations in ORF28 and ORF29 did not affect virus growth but deletions did.

Single genus-specific genes contributed to virus viability, and simultaneous inactivation of several genus-specific genes could give FAdV-4 a replication-defective phenotype. For example, deletion of ORF19A made the FAdV-4 mutant form larger plaques, while recombinant FAdV-4 carrying a deletion of ORF4, ORF12, or GAM-1 produced smaller plaques on LMH monolayers (Fig. 5 and 6) (24). It was reported that GAM-1 had an E1B19K-like function of preventing the host cell from an early apoptosis (30). ORF22 together with GAM-1 played an E1A-like function in regulating the expression of cellular genes to drive the infected cell to enter the S-phase (29). It was reasonable that combined deletions of ORF22 and GAM-1 further reduced the plaque size (Fig. 6B). Deletions of BsiWI-EagI or EagI-SpeI regions at the right end made FAdV-4 replication-defective, which might result from the simultaneous inactivation of several genes including GAM-1 due to the deletion of related promoters or introns. GAM-1-deleted FAdV-4 (XGAM1-CX19A) could grow in LMH cells and primary chicken hepatocytes, but the viability was already compromised, especially in primary cells. Its amplification in chicken embryos was very inefficient (Fig. 11), suggesting that FAdV-4 was less dependent on GAM-1 for replication *in vitro* than *in vivo*. The mechanism behind this deserves further study.

Some helpful information can be deduced from this study for the construction of FAdV-4 as a gene transfer vector. (i) Vector capacity for a cloning transgene can be extended. FAdV4-GX19A-backboned vectors have been routinely used in the laboratory, which carried deletions of 2157 bp at the left and 2420 bp at the right ends of the genome (unpublished data). The deletion of the left end can increase to 3377 bp without compromising the virus growth, e.g., in XBA-GX19A. More combined deletions deserve to be

tested for this aim. (ii) Deletion of genus-specific genes can enhance the growth of recombinant FAdV-4, and such modifications will help increase the vector production. Here, we found that deletion of the EcoRV-XbaI fragment (ORF20) at the right end saved the growth advantage of XEX-CX19A (Fig. 4). (iii) It is possible to develop an attenuated FAdV-4 vaccine. Chicken embryos survived the inoculation of XGAM1-CX19A, and only mild virus replication was detected in the liver (Fig. 11). It is reasonable to believe that XGAM1-CX19A can serve as an attenuated vaccine in young chickens, since chickens possess a more developed immune system than embryos (35). The virulence of FAdV-4 with other deletions also deserves further evaluation in young chickens. (iv) Replication-defective FAdV-4 vectors can be constructed by simultaneously deleting several genus-specific genes. On the other hand, the packaging cell lines for these vectors can be established by exogenously expressing two or more complementary viral genes (36–38).

In conclusion, reverse genetics approaches were employed to construct 21 virus mutants carrying deletions covering the whole left and right ends of the FAdV-4 genome, and it was found that no genus-specific gene was indispensable for virus replication in LMH cells or primary chicken hepatocytes. Our work laid a solid foundation for FAdV-4 vector construction as well as vaccine development.

## MATERIALS AND METHODS

**Plasmids, primers, and reagents for molecular cloning.** pKFAV4GFP, pKFAV4-CX19A, pKFAV4M, and pKFAV7087-Che were constructed in the laboratory previously (24). pKFAV4GFP was a FAdV-4 adenoviral plasmid in which the sequence of ORF1, ORF1B, and ORF2 in the FAdV-4 genome was replaced with CMVp-controlled GFP expression cassette. ORF19A CDS was deleted and GFP CDS was replaced with that of mCherry in pKFAV4GFP to generate pKFAV4-CX19A. pKFAV4M was an infectious clone of FAdV-4 with AvrII site in fiber2 gene mutated synonymously. pKFAV7087-Che was an intermediate plasmid carrying the sequence from SpeI to AvrII sites in pKFAV4-CX19A.

PCR was routinely performed for gene cloning (Q5 High-Fidelity DNA polymerase, Cat. M0491S, New England Biolabs, Ipswich, MA, USA) or plasmid identification before sequencing (Premix Taq, Cat. RR901A, TaKaRa, Dalian, Liaoning, China). DNA recovery and cleaning were performed using kits from Zymo Research (Cat. D4045 and D4010; Irvine, CA, USA). Plasmid construction was conducted using Gibson assembly (NEBuilder HiFi DNA Assembly Master Mix, Cat. E2621; New England Biolabs) or restriction-ligation cloning (DNA Ligation Kit, Cat. 6022Q; TaKaRa) (24, 26, 39). Q5 High-Fidelity DNA polymerase was also used to blunt DNA ends. Restriction enzymes were purchased from New England Biolabs or TaKaRa. Plasmid transformation was performed on *Escherichia coli* TOP10 chemically competent cells with the heat shock procedure according to the manufacturer's instructions (TIANGEN Biotech, Beijing, China). PCR related information and the details of plasmid cloning will be available upon request.

**Cell culture, transfection, and infection.** Chicken hepatoma LMH cells (Leghorn Male Hepatoma, CRL-2117) were purchased from American Type Culture Collection (ATCC, Manassas, VA, USA) and were maintained in Dulbecco's modified Eagle's medium (DMEM) plus 10% fetal bovine serum (FBS; HyClone, Logan, UT, USA) at 37°C in a humidified atmosphere supplemented with 5% $CO_2$, and passaged twice a week. Flasks or plates for cultivating LMH cells were precoated with 0.1% gelatin (Cat. G9391, Sigma-Aldrich, St. Louis, MO, USA) to help cells attach and spread according to the instructions of ATCC. Cells were 1:2.5 split the day before transfection or viral infection. When the culture reached 80% confluence, cells were transfected with plasmid DNA mixed with jetPRIME reagent according to the manufacturer's instructions (Cat. 114-15, Polyplus-transfection, Illkirch, France), or infected with recombinant virus. For the helper plasmid included transfection, 4.5 $\mu$g linearized adenoviral plasmid and 1.5 $\mu$g helper plasmid (pcDNA3-GAM1) were mixed and used to transfect LMH cells in a T25 flask.

**Rescue, purification, and titration of recombinant viruses.** Adenoviral plasmid was linearized by PmeI digestion, and recovered and transfected to LMH cells seeded in T25 flasks. Expression of mCherry or GFP was observed under fluorescence microscope every day posttransfection. If fluorescence foci were found and they grew, the cells together with the culture medium were harvested 5 to 7 days posttransfection when complete CPE occurred, subjected into 3 rounds of freeze-and-thaw, and centrifuged to remove cellular debris. The seed virus was passaged 2 or 3 more times in LMH cells to enrich sufficient progeny viruses for the following experiments. Most FAdV-4 mutants (13/18) were purified with the traditional ultracentrifugation method except that 10 mM citrate (pH 6.2) instead of 10 mM Tris-Cl (pH 7.6) was used as the buffer medium (24, 40). For the purified viruses, particle titer of purified virus was determined by measuring the content of genomic DNA where 100 ng of genomic DNA is equivalent to $2.3 \times 10^9$ viral particles (vp), since a 43 kb genome has a molecular mass of $2.6 \times 10^7$. For all purified or unpurified virus stocks, infectivity titer was determined on LMH cells by the limiting dilution assay in which mCherry+ or GFP+ cells were counted 30 hpi (41). Virus genomic DNA was extracted from purified virions or from virus infected LMH cells by using the modified Hirt's method (42), subjected to restriction analysis, and used as the template for amplifying mutated regions by PCR. PCR products were recovered from agarose gel after electrophoresis and sequenced to confirm the mutation sites. If fluorescence foci could not be found or they were small and could hardly grow after linearized adenoviral plasmid transfection, the detached cells together with the medium were transferred to a 15-mL tube and

collected in 0.4 mL remaining culture medium after centrifugation (100 $g \times$ 10 min) at day 3 and day 5 posttransfection. The cells, together with culture medium in the T25 flask, were harvested at day 7 post-transfection. After 3 rounds of freeze-and-thaw, the rescued viruses were titrated on LMH cells (40, 41).

**Detection of GAM-1 expression by Western blotting.** LMH cells in 6-well plates were transfected with pcDNA3-GAM1 or control pcDNA3 plasmid; 2 days posttransfection, cells were lysed in RIPA lysis buffer (Cat. P0013B, Beyotime Biotechnology, Shanghai, China) and used as the samples for Western blots; rabbit anti-GAM1 antisera were prepared by immunizing rabbits with purified His-tagged prokaryotically expressed GAM-1 protein (Zoonbio Biotechnology, Nanjing, China), 1:50 diluted in PBST (10 mM PBS, 0.05% Tween 20) containing 5% skimmed milk, pre-adsorbed against the nitrocellulose membrane transferred with protein extracted from pcDNA3-transfected LMH cells for 2 h to reduce non-specific antibodies, and then used as the primary antibody for GAM-1 detection; the bands were developed by covering the membrane with Amersham ECL Prime Western Blotting Detection Reagent (Cat. RPN2232, GE, Buckinghamshire, UK) and photographed; the membrane was treated with stripping buffer (Cat. P0025B, Beyotime) after GAM-1 detection; and human $\beta$-actin was stained as a protein-loading control with mouse anti-$\beta$-actin monoclonal antibody (Cat. TA-09, ZSGB-BIO, Beijing, China).

**Plaque forming experiment on LMH cells.** LMH cells in 6-well plates were infected with recombinant virus of 100 infectious units (IU) in 1.5 mL DMEM containing 2% FBS for 2 h. Virus diluent was discarded, and cells were washed twice with PBS (10 mM) and covered with 2.5 mL DMEM containing 2% FBS and 0.8% low-melting agarose (SeaPlaque agarose, Cat. 50100, BioWhittaker Molecular Applications, Rockland, ME, USA) (40). After 4 days' culture, 2 mL fresh liquid DMEM plus 2% FBS was supplemented to each well without disturbance of the semi-solid layer. After 7 days' culture, liquid culture medium was removed carefully, and 2.5 mL 4% paraformaldehyde in PBS was added to the top of semi-solid medium in each well to fix cells. Cells were subsequently stained with crystal violet solution (43), and plaques were photographed using a digital camera. In the late phase of this study, the plaque forming assay was modified in cases when some FAdV-4 mutants could only form small plaques and could hardly be seen with naked eyes. Fluorescence foci instead of plaques were photographed under a fluorescence microscope with a 4× objective lens mounted 5 or 6 days postinfection. The area of all plaques or foci in one or two replicate wells in the culture plate was measured by using the Fiji image processing package (http://fiji.sc/) (44). FAdV4-GFP was included as the control in all batches of plaque forming experiments, and the area of each plaque or focus formed by FAdV-4 mutants was normalized by the median area value of that formed by FAdV4-GFP. The sizes of the plaques were compared to that formed by FAdV4-GFP using the Mann-Whitney nonparametric test.

**Plaque forming experiment on primary chicken embryo hepatocytes.** Specific pathogen free (SPF) chicken eggs were purchased from Beijing Boehringer Ingelheim Vital Biotechnology Company (Beijing, China). Twenty 18-day-old embryos were killed by decapitation. Livers were removed, pooled, weighed, rinsed with cold Hank's Balanced Salt Solutions (HBSS, Cat. SH30268, HyClone), minced, and digested in collagenase (0.5 mg/mL; Cat. 17104019, Thermo Fisher Scientific, Waltham, MA) on a rocker at 37°C for 30 min. The digested sample was filtered through nylon cell strainers (mesh size, 70 $\mu$m; Cat. 087712, Thermo Fisher Scientific) and mixed with an equal volume of 2% bovine serum albumin in HBSS. The filtrate was centrifuged at 300 g for 5 min, then resuspended in 50 mL DMEM supplemented with penicillin-streptomycin (100 U/mL, 100 $\mu$g/mL; Cat. 15140122, Thermo Fisher Scientific). The cell suspension was mixed with an equal volume of 10% sucrose (0.25 M) in Percoll (Cat. P8370, Solarbio, Beijing, China). After centrifugation at 50 g for 10 min, the hepatocytes at the top layer of the Percoll/sucrose were pipetted and transferred to a new 50-mL tube. The Percoll/sucrose density centrifugation was repeated once to remove more contaminated erythrocytes. Isolated hepatocytes were treated with DNase I (working concentration: 25 $\mu$g/mL; Cat. 10104159001, Roche, Mannheim, Germany) in 40 mL DMEM with gently shaking at 37°C until clumps of DNA were no longer visible (10 min) (45). The cells were rinsed twice with DMEM, resuspended in DMEM containing 10% FBS and penicillin-streptomycin, counted with a hemocytometer, and seeded in 6-well plates with a density of $6 \times 10^6$ cells/well. After 40 h cultivation, cells were infected with recombinant FAdV-4 at concentrations of 50, 200, and 500 IU/well for 2 h; the virus-containing media were aspirated; and cells in each well were washed once with DMEM plus 1% FBS and covered with 3 mL DMEM containing 5% FBS, 1% low-melting agarose and penicillin-streptomycin. GFP or mCherry foci were photographed after 6 days' culture, and data were processed as mentioned above.

**Promoter activity.** The sequence from the end of ORF43 to the start of GAM-1 (right-end promoter 1, RP1) was cloned and fused with GFP and mCherry on both sides; the poly(A) signals from SV40 virus and bovine growth hormone (BGH) were added to the terminals of the reporter genes, respectively; and overlap extension PCR products of these elements (mCherry-RP1-GFP) were ligated to plasmid backbone (Kan-Ori) by DNA assembly to generate plasmid pKFAV4RP1-CG. The sequence from the end of GAM-1 to the start of ORF19A (RP2) was cloned and fused with reporter genes similarly to generate plasmid pKFAV4RP2-CG (the details of plasmid cloning will be available upon request). Plasmids pLEGFP-C1 and pmCherry-N1 (Clontech, CA, USA), which carried CMVp controlled GFP or mCherry, served as positive controls. Plasmids were transfected to LMH cells. The expression of reporter genes was firstly observed and photographed under a fluorescence microscope 2 days posttransfection. After that, the cells were detached by trypsin treatment, dispersed into single cells, and suspended in PBS containing 1% FBS and 1.5% paraformaldehyde; the expression of reporter genes was further determined by flow cytometry assay. GFP and mCherry were excited with the 488 nm or 561 nm lasers, respectively (BD LSRFortessa Cell Analyzer, BD Bioscience, CA, USA). The geometric means of fluorescence intensity of total cells were normalized by subtracting the background value of untransfected cells and used to represent promoter activities. The ratios of normalized fluorescence intensities between test and control groups were calculated and considered as the relative promoter activities.

**RNA-seq experiments.** LMH cells in T25 flasks were infected with FAdV4-GFP or FAdV4-CX19A at an MOI of 400 vp/cell for 2 h. Virus was removed, and the cells were washed twice with PBS and cultured in DMEM plus 2% FBS. At 8, 12, or 18 hpi (calculated from the addition of viruses), the culture media were

aspirated and 1 mL TRIzol reagent (Cat. 15596-018, Invitrogen) was added to each flask. The triplicate cell lysates were temporarily preserved at −80°C and later transferred to a company that provided RNA-seq technical service (BGI-Shenzhen, Shenzhen, China). After poly (A) mRNA enrichment, the samples were subjected to nonstranded 150 bp paired-end sequencing. Adapter sequences or low-quality sequences were filtered, and the clean reads were released to the laboratory. HISAT2 aligner was used to map the RNA-seq data to the FAdV4-GFP or FAdV4-CX19A genomes (46); SAMtools were used to convert HISAT2 outputs in SAM format to sorted BAM files and to generate the corresponding index files (47); and the Integrative Genomics Viewer (IGV) was used to view the alignments (48). The coverage and splice junction tracks were displayed and exported as image files.

**Promoter prediction and annotation.** The RP1 sequence was analyzed using the Neural Network Promoter Prediction (NNPP) (49), and a core promoter sequence (gttcaacctatataggtagaccaggtaggcaggttca gacAgacagaccg) was presented with a score of 0.93. The ElemeNT program confirmed the prediction, and further defined "tatatagg" as the TATA box and "tcagaca" as the initiator (50). Sequence ranging from −450 to +50 nucleotides (nt) relative to the predicted +1 nt transcription start site (TSS) was further analyzed with LASAGNA-Search program to search for possible transcription factor binding sites (TFBSs) (51). All 146 JASPAR CORE matrices of vertebrates were selected as the transcription factor (TF) model input, and the cutoff $P$ value was set to 0.001 in the program. The predicted TFBSs and the corresponding transcription factors were further confirmed by using the function of transcription factor DNA-binding matrix searching and scanning in the JASPAR website (52).

**Viral inoculation of embryonated chicken eggs.** Sixty 6-day-old eggs were randomly divided into 4 groups (15 eggs per group): three groups were inoculated with purified FAdV4-GFP, XHE-CX19A, or XGAM1-CX19A of $1 \times 10^8$ vp in 100 $\mu$L PBS via the yolk sac route, and the last group was inoculated with 100 $\mu$L PBS containing no virus and served as noninfected control. All eggs were incubated at 37°C. The viability of the embryos was checked every 24 h. Embryos that died within 48 h postinoculation were treated as non-virus-related death and excluded from the experiment (one death occurred in each of the groups of XHE-CX19A, XGAM1-CX19A, and PBS within 24 h postinoculation). The study endpoint was set to the embryo age of 20 days (14 days postinoculation). After that, the viable embryos were killed by chilling the eggs at 4°C overnight. Livers from dead embryos or those that survived the experiment endpoint were dissected, weighed, minced, and suspended in PBS and frozen at −80°C. After three cycles of freeze-and-thaw, the liver suspension was spun at $1200 \times g$ for 5 min, and the supernatant was titrated on LMH cells by the limiting dilution assay. The yield of virus was normalized by the weight of the liver and presented as infectious units per gram of liver (IU/g) (53–55). The data of viral yields were logarithmically transformed and tested with one-way analysis of variance (ANOVA). The results of embryonic lethality were subjected to survival analysis.

**Data availability.** The RNA-seq data have been mapped to the FAdV4-CX19A genome and deposited at NCBI SRA with the BioProject accession number PRJNA805034.

## ACKNOWLEDGMENTS

This work was supported by the National Natural Science Foundation of China (82161138001, Z.L.). The funders played no role in the study design, data collection and analysis, decision to publish, or preparation of the manuscript.

We declare that we have no conflicts of interest.

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
