## [Reviewer comments · Microbiology Spectrum]

Microbiology Spectrum

No genus-specific gene is essential for the replication of fowl adenovirus 4 in chicken LMH cells

Xinglong Liu, Xiaohui Zou, Wenfeng Zhang, Xiaojuan Guo, Min Wang, Yingtao Lv, Tao Hung, and Zhuozhuang Lu

Corresponding Author(s): Zhuozhuang Lu, National Institute for Viral Disease Control and Prevention, China CDC

Review Timeline:

Submission Date:	February 10, 2022
Editorial Decision:	April 5, 2022
Revision Received:	April 26, 2022
Accepted:	April 28, 2022

Editor: Peter Pelka

Reviewer(s): The reviewers have opted to remain anonymous.

Transaction Report:

DOI: <https://doi.org/10.1128/spectrum.00470-22>

April 5, 2022

Dr. Zhuozhuang Lu
National Institute for Viral Disease Control and Prevention, China CDC
Beijing 100052
China

Re: Spectrum00470-22 (No single genus-specific gene is essential for the replication of fowl adenovirus 4 in chicken LMH cells)

Dear Dr. Zhuozhuang Lu:

I have now received expert reviewer comments, which I have appended below. The reviewers found the manuscript of interest and worthy of publication, however, one reviewer had some issues that I would like you to address and resubmit a modified version.

Link Not Available

Sincerely,

Peter Pelka

Journals Department
Reviewer comments:

Reviewer #1 (Comments for the Author):

Fowl adenovirus (FAdV) is a significant problem in the poultry industry and may be developed as a viral vector. A comparison of the genomes of human adenovirus (HAdV) and FAdV shows that viral proteins involved in replications and structural/assembly aspects of infection are conserved and generally located in the central region of the FAdV genome, while FAdV genus-specific genes are generally located in the unique left and right ends of the viral genome. The authors use reverse genetics to extensively characterize FAdV genus-specific genes (22 designated ORFs). Their results demonstrate these 22 ORFs are dispensable for the replication of FAdV-4 in chicken hepatoma LMH cells and primary chicken embryo hepatocytes. GAM-1 (ORF8)-deleted FAdV-4 formed small plaques and deletion of GAM-1 together with ORF22 resulted in further reduced growth in

LMH cells. Deletion of ORF28, ORF29 and GAM-1 resulted in a defective FAdV-4. A GAM-1-deleted FAdV-4 that replicated efficiently in LMH cells did not kill chicken embryos *in vivo*. These results showed that no one specific gene was essential for FAdV-4 growth and that some genes may have overlapping functions that compensate for one another. Further, the results showed that the effects of specific mutations are context-specific depending on the cells used for assays. Finally, the authors used RNA-seq to annotate gene expression patterns in the genus-specific regions and used reporter assays to identify novel promoter regions in these areas.

This is a comprehensive and convincing report that sheds significant new light on genus-specific genes of FAdV-4. The analyses are conducted in a controlled and technically correct manner. The data are validated using appropriate statistical analyses. The manuscript is very well written and accessible to a diverse audience. Overall, I find this study of general interest to the readership and I have no specific concerns that require revision.

Reviewer #2 (Comments for the Author):

The authors introduced numerous deletions into the left and right end of FAdV genome and they conclude that not a single gene located on those genus specific areas is essential for growth in LMH cells. Altogether, the authors report "negative results" as they didn't find a single gene driving replication in LMH cells which seems a very ambitious hypothesis. Overall, the main outcome seems limited, it indicates that each gene be replaced as no clear adjunct with growth in either LMH or primary cells exists.

Major concerns:

Viruses deleted at the left end of genome carry a reported gene substituting ORF1-ORF-1B and ORF-2 which are obviously not needed for growth and plaque purification. However, from figure 1 it seems that the majority of recombinants contain larger parts being deleted not solely restricted to a single gene.

It seems that rescued viruses were not passaged which does not allow to conclude on stability of the noticed phenotypic trait (lines 128-129 and 452-499). This is of special importance as obviously non-robust growth was seen in some cases (line 123). A more detailed description would be needed.

No scaling of the y-axis (μm) is given in figures.

Fig 5: why was growth of M2829-CX19A already assessed at 3dpi in comparison to all others?

Fig. 11 B: days after inoculation should be given, similar to 11A; why are numbers of eggs for which virus titer was determined different and not always 15?

Lines 349: the explanation for the reduced growth of XGAM1-CX19A is not sound as embryos were infected at 6 days of incubation with nor or very limited impact of the immune system, not comparable with *in ovo* vaccination at day 18. Overall, this would have been a general feature for all mutants. In addition, growth of XGAM1-CX19A on LHM cells seems more rapid than XHE-CX198 (see figures 3 and 5).

Further comments:

Line 62: a more actual version of the reference should be used: Benkő et al. 2022 (J.Gen:Virol.)

Line 79: Griffin et al. 2021 should be referenced

Line 285: viruses from livers of embryos

Staff Comments:

Preparing Revision Guidelines

Please return the manuscript within 60 days; if you cannot complete the modification within this time period, please contact me. If you do not wish to modify the manuscript and prefer to submit it to another journal, please notify me of your decision immediately so that the manuscript may be formally withdrawn from consideration by Microbiology Spectrum.

Response to the first reviewer expert:

The professor had no negative comments.

Response to the second reviewer expert:

1. Viruses deleted at the left end of genome carry a reported gene substituting ORF1-ORF-1B and ORF-2 which are obviously not needed for growth and plaque purification. However, from figure 1 it seems that the majority of recombinants contain larger parts being deleted not solely restricted to a single gene.

What the reviewer professor pointed out is true. We explained this in the first subsection of “The strategy of constructing FAdV-4 mutants” in Results: The restriction site-defined regions of the viral genome were systematically deleted to detect the key regions for virus replication. Frameshift mutations or coding sequence (CDS) deletions were further introduced into the key regions to distinguish the effects of individual genes in the following steps.

After carefully considering the professor’s comment, we changed the title to "No genus-specific gene is essential for the replication of fowl adenovirus 4 in chicken LMH cells". Such modification could make it more stringent. In the revised manuscript, we deleted the description of "single gene" in some places while reserve it in others where the term of "single genes" were used to contrast with combined several genus-specific genes.

2. It seems that rescued viruses were not passaged which does not allow to conclude on stability of the noticed phenotypic trait (lines 128-129 and 452-499). This is of special importance as obviously non-robust growth was seen in some cases (line 123). A more detailed description would be needed.

For the first version of this manuscript, we only studied several FAdV-4 mutants with deletions at the right end. In that situation, all these mutants were purified. After

that, we constructed more FAdV-4 mutants, and finally 13 of the total 18 rescued mutants were purified. Some FAdV-4 mutants were not purified because we thought purification was not necessary. For the unpurified viruses, only infectious titers were determined. All these rescued viruses were passaged at least 3 times on LMH cells to enrich sufficient progeny viruses for the following experiments. For these rescued viruses, plaques were formed and CPE occurred when linearized adenoviral plasmid was used to transfect LMH cells, suggesting an acceptable growth rate. When the seed viruses were amplified in LMH cells, the propagation of progeny viruses was sustained and stable. The viruses were identified by restriction analysis of the genomic DNA and by sequencing the mutated regions after being passaged 3 or 4 times in LMH cells.

Lines 128-129 were changed to: Plaque forming experiments were carried out to compare the viability of rescued FAdV-4 mutants.

Line 453: We deleted the description of 40,000 vp in the subsection of “Plaque forming experiment on LMH cells” for consistency since 40,000 vp is approximately equivalent to 100 IU.

Lines 422-428: A more detailed description was added to the subsection of “Rescue, purification and titration of recombinant viruses” in Materials and Methods:

The seed virus was passaged 2 or 3 more times in LMH cells to enrich sufficient progeny viruses for the following experiments. Most FAdV-4 mutants (13/18) were amplified in LMH cells, purified with the traditional ultracentrifugation method.....

For all purified or unpurified virus stocks, infectivity titer was determined on LMH cells by the limiting dilution assay in which mCherry+ or GFP+ cells were counted 30 hpi.

3. No scaling of the y-axis (μm) is given in figures.

For the cell images, the scale bars are identical for both x- and y-axes. We think it could be appropriate to label one scale bar for the whole figure.

4. Fig 5: why was growth of M2829-CX19A already assessed at 3dpi in comparison to all others?

M2829-CX19A grew faster than all other FAdV-4 mutants in this figure. At day 3 post transfection, the rescued viruses already spread to all cells. At day 5, CPE occurred. We thought that the rapid growth of M2829-CX19A could be illustrated more clearly by providing the cell image at days 2 and 3 post transfection.

5. Fig. 11 B: days after inoculation should be given, similar to 11A; why are numbers of eggs for which virus titer was determined different and not always 15?

We did the modification as the professor suggested. To titrate virus from so many livers is very laborious. For these who survived the observation endpoint, livers from 6 embryos were randomly selected for virus titration because such number of samples could provide enough data for statistical analysis.

6. Lines 349: the explanation for the reduced growth of XGAM1-CX19A is not sound as embryos were infected at 6 days of incubation with nor or very limited impact of the immune system, not comparable with in ovo vaccination at day 18. Overall, this would have been a general feature for all mutants. In addition,

growth of XGAM1-CX19A on LHM cells seems more rapid than XHE-CX198 (see figures 3 and 5).

We agree with the professor. Although cells for innate immune system can be detected in 4-day-old chicken embryos [pubmed:31063007], there is no evidence that they can protect the embryos from virus infection. We searched and read some publications. However, we could not find pertinent knowledges to describe the difference of virus growth in cultured cells and in early chicken embryos. We choose to delete this sentence.

XHE-CX19A and XGAM1-CX19A seemed to have similar growth rate in LMH cells. Both propagated slower than FAdV4-GFP although the difference between XHE-CX19A and FAdV4-GFP was not statistically significant (Figure 4 and Figure 6). In contrast, in primary chicken embryo hepatocytes, XHE-CX19A and FAdV4-GFP had similar growth rate while the propagation of XGAM1-CX19A was much slower (Figure 7). The results of virus growth in primary hepatocytes were consistent with the data from chicken embryo inoculation experiments. We re-wrote this paragraph as following:

GAM-1-deleted FAdV-4 (XGAM1-CX19A) could grow in LMH cells and primary chicken hepatocytes, but the viability was already compromised, especially in primary cells. Its amplification in chicken embryos was very inefficient (Fig 11), suggesting that FAdV-4 was less dependent on GAM-1 for replication in vitro than in vivo. The mechanism behind this deserves further study.

7. Line 62: a more actual version of the reference should be used: Benkő et al. 2022 (J.Gen:Virol.)

We updated the taxonomy of adenoviridae and cited the latest publication in the revised version. Thanks.

8. Line 79: Griffin et al. 2021 should be referenced

We cited this publication in the revised manuscript.

9. Line 285: viruses from livers of embryos

We corrected this mistake as the professor suggested.

April 28, 2022

Dr. Zhuozhuang Lu
National Institute for Viral Disease Control and Prevention, China CDC
Beijing 100052
China

Re: Spectrum00470-22R1 (No genus-specific gene is essential for the replication of fowl adenovirus 4 in chicken LMH cells)

Dear Dr. Zhuozhuang Lu:

Your manuscript has been accepted, and I am forwarding it to the ASM Journals Department for publication. You will be notified when your proofs are ready to be viewed.

Sincerely,

Peter Pelka
Editor, Microbiology Spectrum
